# Study protocol for the 'preventing functional decline in acutely hospitalised older patients (PREV_FUNC)' study: effects of two multicomponent exercise programmes on physical function – a three-armed randomised controlled trial

Anna-Karin Welmer [1,2] Linda Sandberg, [1,3] Christina Sandlund, [1,4] Caroline Björck, [5,6,7] Maria Hagströmer, [1,4] Julia Hamilton, [8] Gunilla Helgstrand, [1,8] Charlotte Lindgren, [3] Linda Nordstrand, [1,2] Petter Sandstedt, [2] Miia Kivipelto, [1,9,10] Anne-Marie Boström [1,9,10]

For numbered affiliations see end of article.

**Correspondence to**
Dr Anna-Karin Welmer;
anna-karin.welmer@ki.se

## ABSTRACT

**Introduction** Acutely hospitalised older patients often live with frailty and have an increased risk of impaired physical function. Previous studies suggest that exercise might mitigate the risk of physical impairment; however, further research is needed to compare the effect of different types of exercise interventions. In this paper, we report a protocol for a trial that aims to examine (1) if multicomponent exercise interventions (interventions that include both mobility and strengthening exercises) have effects on physical function compared with usual care in older adults and (2) if a comprehensive multicomponent exercise programme is more effective than a simple multicomponent exercise programme that only include walking and sit-to-stand exercises.

**Methods and analysis** This is a three-armed randomised controlled trial, with two intervention groups (comprehensive and simple exercise programme) and a control group receiving usual care. We will include 320 participants aged ≥75 years from geriatric medical departments of four hospitals in Stockholm, Sweden. Assessments will be conducted at hospital admission, discharge and 3 months thereafter concerning physical function (primary outcome), activities of daily living, health-related quality of life, sarcopenia and falls. The number of readmissions will be registered up to 1 year after discharge. Data will be analysed with linear mixed effects models, according to the intention-to-treat approach.

**Ethics and dissemination** Ethical approval for this trial has been granted by the Swedish Ethical Review Authority (approval number 2022-03032-01). Data collection will consider the information requirement, the requirement of consent, confidentiality obligations and the utilisation requirement. Trial findings will be disseminated through multiple channels, including scientific publications and conferences, and workshops with healthcare professionals and the public.

**Trial registration number** NCT05366075

## STRENGTHS AND LIMITATIONS OF THIS STUDY

⇒ In order for the sample to be representative of older hospitalised patients, the participants will be consecutively recruited at several hospitals in Stockholm.
⇒ Both exercise interventions use none or low technology exercise equipment that is already available in most hospital settings or can be purchased at a low price.
⇒ The block design may result in selection bias in the trial; however, one advantage is that participants from both groups at the same department are avoided.

## INTRODUCTION

Hospitalisation due to acute medical illness is associated with several negative health consequences in older adults, even if the illness that caused the admission is successfully treated. Approximately, half of adults aged 65 years and older show a decline in physical function following acute hospitalisation.[1] Many patients do not regain prehospital function, lose independence and have increased risk of reduced well-being, readmissions and death after discharge.[2 3]

Acutely hospitalised older patients often live with frailty and several concurrent diseases.[3] Furthermore, acutely hospitalised

older people spend most of their hospital stay in bed, even if they are able to walk independently.[4] Such periods of physical inactivity, although short, may exacerbate the age-related loss of muscle mass, strength and physical function.[5 6] It has been suggested that relatively small amounts of exercise might mitigate these deficits,[7 8] emphasising the importance of identifying exercise interventions that are easy to use for older people during hospital stay.

A recent Cochrane review concluded that there is uncertainty regarding the effect of exercise interventions on functional outcomes for older medical inpatients during acute hospitalisation.[9] Thus, it has been suggested that future research on this topic should focus on consistent reporting of participant characteristics, including baseline level of functional ability, as well as exercise dose and adherence to provide an insight into the reasons for the observed inconsistencies in findings.[9]

In a randomised controlled trial (RCT), Martínez-Velilla et al demonstrated that a comprehensive exercise programme, which consisted of several supervised exercises including progressive resistance, balance and walking exercise, provided significant health benefits over usual care in acutely hospitalised patients aged 75 years and older.[10] Also, in another RCT, Ortiz-Alonso et al showed that a relatively simple multicomponent intervention solely consisting of walking and rising from a chair decreased the risk of dependence in activities of daily living (ADL) in acutely hospitalised patients older than 75 years.[11] It is not known whether a simple exercise programme, such as the one by Ortiz-Alonso et al could yield similar health benefits as a more comprehensive intervention, such as the one by Martínez-Velilla et al. Considering the time constraints in acute care settings, a simple exercise programme may be more feasible if proven effective.

The main aims of the 'preventing functional decline in acutely hospitalised older patients (PREV_FUNC)' study are to (1) evaluate the effect of multicomponent exercise interventions (interventions that include both mobility and strengthening exercises) on physical function (primary outcome), ADL, health-related quality of life (HRQL), sarcopenia, falls, readmissions and mortality in comparison with usual care in acutely hospitalised people aged 75 years and older and (2) if a comprehensive multicomponent exercise programme that include several exercises is more effective than a simple multicomponent exercise programme that only include walking and sit-to-stand exercises. An initial pilot of the PREV_FUNC study suggested that the intervention was feasible, safe and acceptable to the participants (unpublished data). In this paper, we present a study protocol for the PREV_FUNC study.

## METHODS AND ANALYSIS
### Trial design
The PREV_FUNC study will be conducted in accordance with the Consolidated Standards of Reporting Trials.[12] It is a three-armed single blind block randomised controlled multi-centre trial. Participants will be included consecutively from geriatric medical departments at four hospitals in Stockholm, Sweden. A flow chart of the participants is shown in figure 1. The patients will be included in the control group or one of the intervention groups in a time-dependent manner: participants will be recruited to each group during blocks of 2–3 weeks, which will be repeated until the end of the trial. The order of the groups will be randomised for each trial site (randomisation will be done by the responsible researchers). The block design has been chosen to avoid copresence of participants from intervention and control groups at the same department; that is, to prevent the exercises in the intervention groups from spreading to usual treatment and risk contaminating the control group.[11 12] Eligible subjects will be invited to participate in the trial within 36 hours of admission and the baseline examination will be conducted directly after the inclusion. The trial has been registered at Clinical-Trials.gov, NCT05366075. This protocol is reported in accordance with the Standard Protocol Items: Recommendations for Interventional Trials.[13]

### Inclusion and exclusion criteria
Inclusion criteria for participation are age 75 years and older, ability to stand up from a sitting position independently or with minimal personal help, and ability to communicate and collaborate with the research staff. People who cannot follow instructions will be excluded. Patients who by the responsible physician are assessed as not eligible to participate due to terminal illness or any major medical condition that contraindicates exercise will also be excluded. We will also exclude patients who live in nursing homes, or those who were previously included in the trial.

### Interventions
The interventions will start the same day as the baseline examination or the day after and will be performed on all weekdays until discharge. Intervention 1 is adapted from the exercise programme by Ortiz-Alonso et al,[11] and intervention 2 is adapted from the exercise programme by Martinez-Velilla et al.[10] The interventions will be offered in addition to usual care.

#### Intervention 1 (simple exercise programme)
This intervention will include up to four sessions per day (total duration 20–30 min/day). It consists of sit-to stand exercises and walking along the corridor of the ward (for those who can walk). The exercises will mainly be led by physiotherapists, and, when suitable, intervention 1 will be performed in connection with regular routines in the usual care (table 1). At discharge, the participants will be encouraged to continue with the exercise programme at home, 3 days per week during 6 weeks. In addition to oral instructions, they will receive written instructions with pictures on how to perform the exercise.[11]

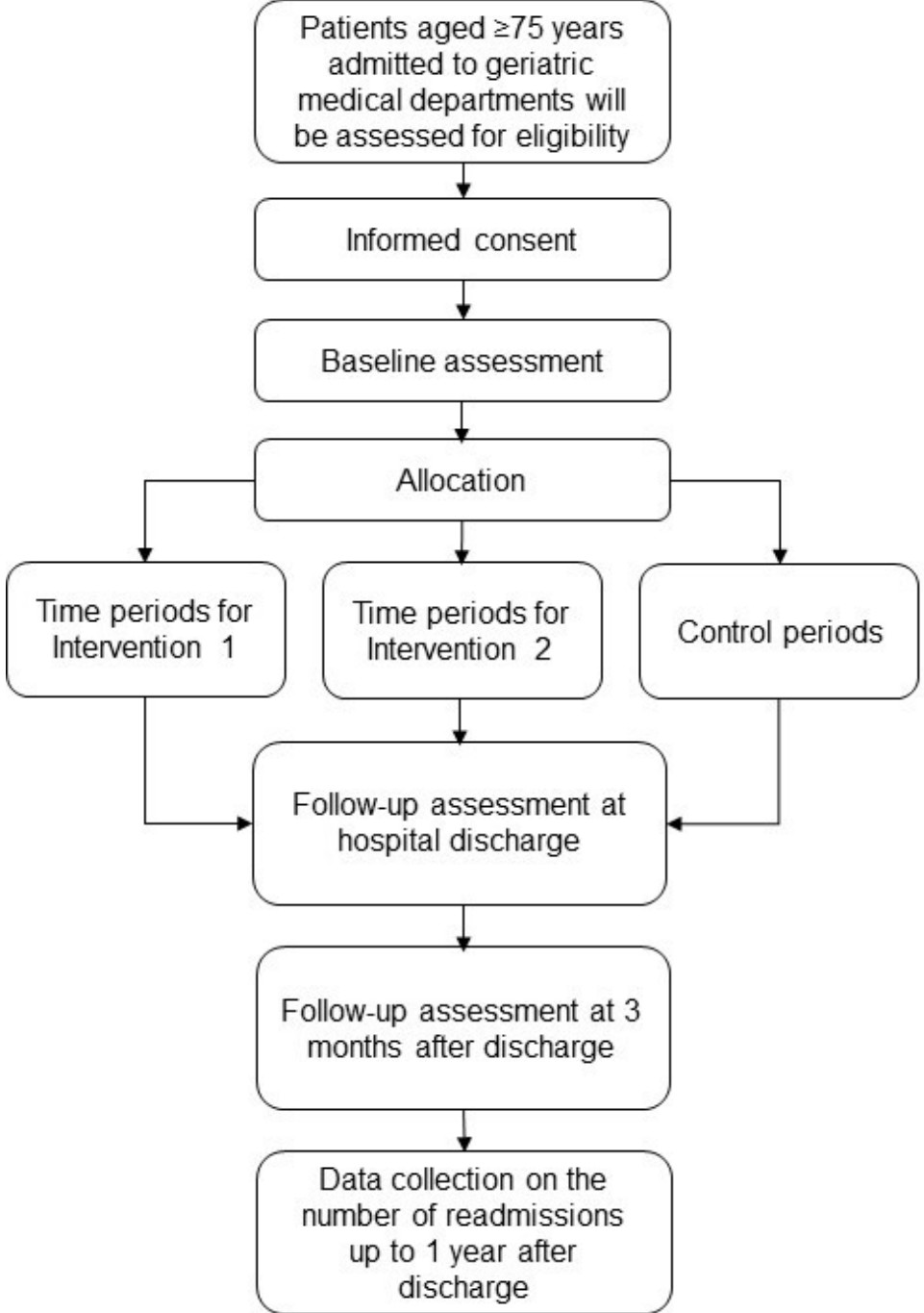

**Figure 1** Flow chart of the participants.

### Intervention 2 (comprehensive exercise programme)

This intervention consists of two daily sessions (morning and afternoon) with a duration of 20 min each. The morning session includes individualised supervised progressive resistance, balance and walking exercise, tailored to each participant's capacity. The resistance exercise includes using weight cuffs, which mainly involves lower-extremity muscles. Balance and gait exercises include different exercises such as line walking, stepping practice and walking with small obstacles. The evening session consists of functional exercises using light loads, such as knee extension and flexion, hand exercise with a ball, and daily walking exercise (for those who can walk).[10] The exercises will be led by physiotherapists (table 1).

### Control group

The control group will receive usual care. Usual care is based on teamwork, which includes physiotherapy. The physiotherapy focuses on mobility assessments, and the patients may receive a limited amount of functional training when needed. However, we expect that the participants often have a limited amount of physical activity during the hospital stay.[14]

**Table 1** Overview of the interventions

| Intervention | Timing/frequency and duration of exercises | Description | Dose |
|---|---|---|---|
| Intervention 1 (Simple exercise programme) | Up to four times/day (total duration 20–30 min/day) | ▶ Sit-to-stand exercises. Rising from a chair and sitting down again repeatedly, using armrest and assistance if needed. | 1–3 sets of up to 10 repetitions, depending on the participant's ability. |
| | | ▶ Walking. Walking along the corridor, using walking aid and assistance if needed. | 3–10 min per session, with resting periods if needed. |
| Intervention 2 (Comprehensive exercise programme) | Before lunch (duration 20 min) | Strengthening exercises<br>▶ Sit-to-stand exercises. Rising from a chair and sitting down again repeatedly, using armrest and assistance if needed.<br>▶ Knee extensions. Knee extensions in a sitting position, with a weight cuff on the ancle. Repeat with both legs.<br>▶ Standing push-ups. Bending and stretching arms leaning on a wall. | 1–3 sets of up to 10 repetitions, depending on the participant's ability. |
| | | Balance exercises<br>▶ Semi-tandem stance. One foot placed in front of the other foot, with the big toe of the back foot in the groove of the front foot. Repeat on the other side.<br>▶ Line walking. Walking in a straight line, placing one foot in front of the other and vice versa.<br>▶ Walking with small obstacles. Walking around small obstacles and/or stepping over small obstacles.<br>▶ Stepping practice. Stepping up and down on a step board or staircase.<br>▶ Weight transfer. Standing up, transferring the weight from one leg to the other.<br>▶ Proprioceptive exercises. Walking on an unstable surface. | Dose adapted to the participant's ability and time frame. |
| | Afternoon (duration 20 min) | ▶ Knee extensions. Knee extensions in a sitting position. Repeat with both legs.<br>▶ Knee flexion. Standing knee flexions, holding on to a chair. Repeat with both legs.<br>▶ Hip abduction. Standing hip abduction with the participant standing, holding on to a chair. Repeat with both legs.<br>▶ Hand exercise with handgrip ball.<br>▶ Walking. Walking along the corridor, using walking aid and assistance if needed. | 1–3 sets of up to 10 repetitions for, depending on the participant's ability. |

Daily logbooks will be used during the hospital stay to register the number of attended exercise sessions for the intervention groups, as well as treatment safety for all groups (by registering adverse events during the hospital stay, eg, falls). Furthermore, exercise diaries will be used during 6 weeks to register the self-training conducted at home in intervention 1.

### Data collection
Physical function, ADL, HRQL, sarcopenia and falls will be assessed at baseline (hospital admission), discharge and 3 months after discharge. Research staff blinded to group allocation will conduct the tests and administer the questionnaires at the hospital (for the baseline and discharge evaluations) and at the home of the participants (for the 3-month evaluation). The 6-month and the 1-year evaluation will be conducted using patient records (for hospital readmissions and mortality) (table 2).

The primary outcome will be lower extremity physical function, assessed according to the Short

**Table 2** Time of assessment of the different variables in the trial

| Assessment | Baseline (hospital admission) | After exercise/control period (discharge) | 3 months after discharge | 6 months after discharge | 1 year after discharge |
|---|---|---|---|---|---|
| **Time point** | | | | | |
| Primary outcome | | | | | |
| Physical function | X | X | X | | |
| Secondary outcomes | | | | | |
| Activities of daily living | X | X | X | | |
| Health-related quality of life | X | X | X | | |
| Sarcopenia | X | X | X | | |
| Falls after inclusion | | X | X | | |
| Hospital readmissions | | | X | X | X |
| Mortality | | | X | X | X |
| Additional data | | | | | |
| Frailty status | X | | | | |
| Nutritional risk | X | | | | |
| Nutritional supplementation | Continuously collected during the hospitalisation | | | | |
| Falls in the previous year | X | | | | |
| Demographic data | X | | | | |
| Reason for hospital admission | X | | | | |
| Comorbidities | X | | | | |
| Length of hospital stay | | X | | | |
| Discharge destination | | X | | | |

Physical Performance Battery (SPPB).[15] The SPPB has three components: standing balance, walking speed and the chair stand test for lower extremity strength. It has been validated in older adults living with frailty, is sensitive to change, and predicts a wide range of clinical outcomes.[16 17]

Secondary outcomes will be ADL (according to the Barthel Index),[18] HRQL (according to the EuroQol-5 Dimension),[19] presence of sarcopenia (defined by handgrip strength, calf circumference and arm circumference), according to the European Working Group's revised criteria on Sarcopenia in Older People,[20] and number of falls during follow-up (according to the logbooks at the hospital, and self-report at the 3-month follow-up). Data on vital status and number of readmissions up to 1 year after discharge will be collected using patient records.

Baseline data will also be collected from patient records including demographic data (age, sex and cohabitation status), reason for hospital admission, comorbidities, frailty status and nutritional risk. Disease diagnoses (reason for hospital admission and comorbidities) will be coded based on the International Classification of Diseases 10th Revision, and further operationalised using a previously described methodology.[21] Frailty status will be defined according to the Clinical Frailty Scale,[22] and nutritional risk will be defined according to the Mini Nutritional Assessment-Short Form.[23] Type and amount of nutritional supplementation consumed during the hospitalisation (if applicable) will be recorded in a logbook. Falls in the previous year will be assessed according to self-report at baseline. Length of hospital stay and discharge destination will be registered at discharge.

## Data analysis
### Power of the trial
We estimate that 264 participants should be included in the main RCT to enable detection of a clinically meaningful difference of one point in the main outcome (SPPB) at 3 months after discharge,[24] when power is 0.80 and alpha is 0.05 (three group comparison). To take into account, approximately 20% of deaths or dropouts during 3 months, 320 participants should be included in the trial.

We will use the intention-to-treat approach. Between-group comparisons of change in the continuous variables from baseline to follow-up will be conducted using linear mixed effects models, with adjustments for baseline scores. To account for deaths and lost to follow-up, we will model the group comparisons through joint models, which will be performed by simultaneously estimating both a linear mixed model and a Cox's proportional hazard regression model. Group differences in the number of

readmissions will be examined using ordinal logistic regression models. Binary logistic regression models will be employed to examine between group differences in binary outcomes, and Cox's proportional hazard regression model will be used for mortality data. Multiplicative interactions between sex and group allocation in relation to the outcomes will also be examined. Statistical analyses will be performed with Stata and R (version 16).

## Patient and public involvement

The trial is conducted in dialogue between researchers, clinicians and healthcare managers of different disciplines. Although the interventions will mainly be led by physiotherapists, an intervention at a geriatric department requires a dialogue with several professions to meet the needs of the older adults, who often have several concurrent health problems. Thus, the trial has an interdisciplinary and practice-oriented focus, which will facilitate future implementation if the interventions are proven to be effective. Furthermore, qualitative interviews will be conducted among the participants in the two intervention groups, to explore their experiences of the interventions. The interviews will be conducted at the hospital during the hospital stay, or at the participant's home after discharge.

## Trial status

This is version 1 of the study protocol. Recruitment for the trial is ongoing and is planned between September 2022 and December 2023. Any deviations to this protocol will be updated on ClinicalTrials.gov, and the changes will be reported when disseminating the results.

## ETHICS AND DISSEMINATION

Ethical approval for this trial has been granted by the Swedish Ethical Review Authority (approval number 2022-03032-01). Data collection will consider the information requirement, the requirement of consent, confidentiality obligations and the utilisation requirement. The participants will receive verbal and written information, and informed written consent will be collected by research staff (online supplemental file S1). All data collected are pseudonymised and stored in paper and digital format in accordance with the General Data Protection Regulation. Only the responsible researchers will have full access to the data. Trial findings will be disseminated through multiple channels, including scientific publications and conferences, and workshops with healthcare professionals and the public. We will follow the Vancouver recommendations for authorship.

## DISCUSSION

The goal of this trial is to provide knowledge on how to prevent physical impairment in acutely hospitalised patients aged 75 years and older. Although several studies have shown beneficial effects of community-based exercise programmes in older adults, less attention have been given to exercise programmes during hospitalisation for acute medical illness.[25] To date, there is no consensus concerning the most effective exercise intervention to prevent functional decline in acutely hospitalised older adults.[26]

Older adults consume a large proportion of healthcare resources.[27] Current guidelines for care of various diseases are mainly based on research performed on younger people without severe comorbidity and cannot always be generalised to older patients. This trial may provide a novel strategy to maintain or improve physical function in this group of older adults living with frailty. If the trial will show positive effects, we plan to further study its implementation and analyse its cost-effectiveness in subsequent studies. If our results are positive, it might lead to a change in clinical practice where multicomponent exercises are incorporated in routine care. Our outcome, physical function, is an important component of healthy ageing and a strong predictor of future health status and mortality following hospitalisation in older adults.[16 17]

Our interventions are based on two previous studies conducted among Spanish hospitalised older adults.[10 11] The results from these studies indicate that multicomponent exercises are safe and effective in counteracting functional limitations in older hospitalised people. However, research is needed on the generalisability of these approaches to other settings. Furthermore, it is not known whether these approaches are equally efficient in counteracting functional limitations in older hospitalised people. In the present trial, we have modified the intervention by Martínez-Velilla et al,[10] by using weight cuffs instead of resistance exercise machines. This adaption was done to enhance future implementation in clinical settings. The exercise interventions examined in this trial use either no exercise equipment or low technology exercise equipment that is already available in most hospital settings or can be purchased at a low cost. Furthermore, we modified the intervention by Ortiz-Alonso et al by increasing the total duration of the exercise from 20 min per day to 20–30 min per day.[11] This was done to make the two interventions more comparable in terms of daily exercise time during the hospital stay. Finally, the interventions will only be performed on weekdays to fit with staff resources during weekends.

We expect that this trial will help identify the exercise programme that is most effective, and therefore, may be most suitable for future implementation. The two exercise programmes have different possible strengths and weaknesses. For instance, it may be hypothesised that the comprehensive exercise programme may have a better effect on physical function at hospital discharge since it is more individualised and intensive than the simple exercise programme. However, a disadvantage is that it is less suitable for self-training than the simple exercise

programme, due to its complexity and the possible risk entailed by doing unsupervised balance training. Thus, it may be hypothesised that the simple exercise programme, which is more suitable for self-training, can have an equal or better effect on physical function than the comprehensive exercise programme 3 months after discharge since the participants can continue with the programme at home. Furthermore, in future studies, we plan to conduct post hoc analyses to identify associations between baseline factors, for example, frailty level and the effect of the intervention.

This trial has some limitations. First, instead of a classic randomisation, we have chosen a block design, which may introduce selection bias in the trial. However, an advantage of the block design is that it avoids copresence of participants from both groups at the same department; and thus, prevents that the exercises in the intervention groups will spread to ordinary treatment and risk to contaminate the control group. If the exercises in the intervention groups would spread to the control group and produce unexpected clinical benefits, it could create a bias by disguising a possible genuine benefit of the interventions and reduce the effect size to a value that is not clinically significant. Therefore, it has been emphasised that researchers in clinical trials on rehabilitation topics should make sure that control groups do not include any of the active elements of the intervention such as physical exercise.[12] Second, like usual care in the studies by Martínez-Velilla *et al* and Ortiz-Alonso *et al*,[10 11] usual care at geriatric departments in Sweden include physiotherapy, and the patients may receive a limited amount of functional training when needed. This may introduce a bias in the trial since participants in the control group may also receive some physical exercise. However, a hospital stay is often accompanied by low levels of physical activity, which has been suggested to play a major role in causing the negative health consequences associated with hospitalisation.[4 14] Third, the trial will include participants who are able to stand up from a sitting position, and to communicate and collaborate with the research team, which could limit the generalisability to those with relatively higher levels of functioning. However, the trial participants will be consecutively recruited at several hospitals, so that the sample will be representative of older hospitalised patients.

A possible challenge in this trial may be the gradual change towards shortened length of hospital stay for geriatric inpatients in Sweden.[28] A further shortened average length of hospital stay may limit the possibilities to show significant effects of the interventions during the hospital stay. Thus, in intervention 1, the participants will be encouraged to continue with the exercises at home. In future studies, we plan to explore prerequisites for a transitional care intervention, in which physical exercise is performed during acute hospitalisation and as self-training at home with support from primary care. This three-armed intervention is a first step towards finding the best strategies to maintain or improve physical function among acutely hospitalised older adults.

In summary, the PREV_FUNC trial has the potential to provide new knowledge about how to maintain and improve physical function in acutely hospitalised older patients. The number of hospital admissions is likely to increase as the population ages. Identifying the most effective interventions to reduce health consequences of acute hospitalisation in older adults is thus highly relevant from clinical and public health perspectives.

**Author affiliations**
¹Department of Neurobiology, Care Sciences and Society, Karolinska Institutet, Stockholm, Sweden
²Women's Health and Allied Health Professionals Theme, Karolinska University Hospital, Stockholm, Sweden
³Department of Geriatric Medicine, Dalengeriatriken Aleris Närsjukvård AB, Stockholm, Sweden
⁴Academic Primary Health Care Centre, Region Stockholm, Stockholm, Sweden
⁵Department of Women's and Children's Health, Uppsala University, Uppsala, Sweden
⁶Centre for Research and Development, Region Gävleborg, Gävle, Sweden
⁷Department of Caring Sciences, University of Gävle, Gävle, Sweden
⁸Department of Geriatric Medicine, Sabbatsberg Hospital, Stockholm, Sweden
⁹Theme Inflammation and Aging, Karolinska University Hospital, Huddinge, Sweden
¹⁰Research and Development Unit, Stockholm's Sjukhem, Stockholm, Sweden

**Contributors** A-KW, A-MB and LS designed the protocol and methods of the trial. A-KW, LS, CS, CB, MH, JH, GH, CL, LN, PS, MK and A-MB contributed with the writing of the manuscript. All authors read and approved the final manuscript.

**Funding** The PREV_FUNC trial is supported by grants provided by the Swedish Research Council for Health, Working Life and Welfare, Grant Number 2021-01788; Region Stockholm (ALF project), Grant Number FoUI-960460, Center for Innovative Medicine (CIMED), Region Stockholm, Grant Number FoUI-961330; and Konung Gustaf V:s och Drottning Victorias Frimurarestiftelse (PI A-KW). In addition, MK was supported by Stiftelsen Stockholms Sjukhem, CIMED and the Swedish Research Council, Grant Number 2017-0610.

**Disclaimer** The funders have no role in design of the trial, collection, management, analysis, interpretation of data, writing manuscripts, or in the decision to submit for publication.

**Competing interests** None declared.

**Patient and public involvement** Patients and/or the public were involved in the design, or conduct, or reporting, or dissemination plans of this research. Refer to the Methods section for further details.

**Patient consent for publication** Consent obtained directly from patient(s).

**Provenance and peer review** Not commissioned; externally peer reviewed.

and indication of whether changes were made. See: https://creativecommons.org/licenses/by/4.0/.

**ORCID iDs**
Anna-Karin Welmer http://orcid.org/0000-0001-5819-8724
Anne-Marie Boström http://orcid.org/0000-0002-9421-3941

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
