## [Reviewer comments · BMJ Open]

ARTICLE DETAILS

TITLE (PROVISIONAL)	A study protocol for the "Preventing functional decline in acutely hospitalized older patients (PREV_FUNC)" study – Effects of two multicomponent exercise programs on physical function: a three-armed randomized controlled trial
AUTHORS	Welmer, Anna-Karin; Sandberg, Linda; Sandlund, Christina; Björck, Caroline; Hagströmer, Maria; Hamilton, Julia; Helgstrand, Gunilla; Lindgren, Charlotte; Nordstrand, Linda; Sandstedt, Petter; Kivipelto, Miia; Boström, Anne-Marie

VERSION 1 – REVIEW

REVIEWER	O'Shaughnessy, Íde University of Limerick, School of Allied Health
REVIEW RETURNED	11-Mar-2023

GENERAL COMMENTS	Thank you for sharing this protocol for your trial. I understand from the methods that recruitment and data collection commenced in September 2022. The protocol serves to answer an important knowledge deficit and is strengthened by a range of outcome variables. However, I would like the author to consider and address the following revisions: The introduction should reference and highlight key findings from the Cochrane review published in November 2022 "Exercise for acutely hospitalised older medical patients". The authors commentary regarding implications for future research arising from the review, should serve as the justification for this trial. Terminology: the term older adults who are frail cited throughout the paper should be replaced with "older adults living with frailty" Inconsistent language noted throughout paper i.e trial, study, project Please amend the following sentence as the language used, may be perceived as ageist "Acutely hospitalized patients aged 75 years and older often suffer from frailty and multimorbidity and consume a large proportion of the resources of health care (24)" Eligible participants and baseline characteristics: While the authors have outlined inclusion and exclusion criteria for participation, please outline what clinical criteria or measurement will be utilised to determine "severe confusion". Inclusion of socioeconomic variables e.g level of education, previous occupation and a profile of baseline characteristics e.g. Charlson Comorbidity Index should be considered. Use of ICD 10 codes should be considered to standardise recording of "reason for hospital admission".
---

	Randomisation process: Given the risk of selection bias imposed by using the block randomisation design outlined in this protocol, please provide a further an evidence-based justification for using this design. PPI: The PPI element of this protocol appears to be largely focused on healthcare professionals and researchers experiences of an interdisciplinary approach and less about the older adults experiences of recruitment and adherence to the intervention. Therefore, please provide further information on the planned conduct of the "qualitative interviews on the participants' views of the interventions".
--	---

REVIEWER	Han, chad Yixian Flinders University College of Nursing and Health Sciences
REVIEW RETURNED	03-Apr-2023

GENERAL COMMENTS	Thank you for the opportunity to review your protocol paper, this area of research is very important. Thank you for furthering the evidence in this field. My comments are purely from a research and practice point of view and are not intended to malign any organization, or individual. Strength and limitations of this study  • Study participants will.....so the that the sample will be representative of older hospitalised patients in Sweden • While the project has a clear practice-oriented focus, it is not clear that it is interdisciplinary as only exercise (no nutrition or psychological) interventions were tested. • It is known that such interventions cannot be double blinded unless there is a sham intervention. As such, I do not think that there is a need to list that as a key limitation of the study in the key highlights. Introduction  • Acutely hospitalized older patients are often frail... (as opposed to have frailty) • Such periods of physical activity, albeit short, may exacerbate..... • Please replace “ Identifying the most efficient interventions to reduce health consequences of acute hospitalization in older adults should be a high priority in society, especially the increasing older population” with something more specific to the acute care setting. I do get the message that there is ageing population but this sentence is too generic and does not highlight the importance of this study. Suggest to authors to highlight the time constraints in an acute setting and how a simple exercise, if proven effective, can be cost-effective in a time-pressure acute care setting. Methods and analysis  • The study has been registered at clinicaltrials.gov. ◊ Please provide the registration details i.e., number. • Please reference appropriately. For example, the SPIRIT checklist needs to be referenced, along with previous studies that used block design. • Please provide details on how usual care will be documented. We'd expect varied care depending on physician's referral to physiotherapy departments during the stay. • The mini-nutritional assessment tool short form is a screening tool (assess nutritional risk) and cannot provide nutrition assessment. Please change the description where this is discussed within the
--

	manuscript.  • Nutrition supplementation is collected during the hospitalization. Any physical activity in standard care should also be documented too because standard care varies between hospitals Discussion  • Patients in intervention 1 will be encouraged to continue with the exercises at home but not intervention 2? • While I agree that measuring effectiveness is important, we know that compliance is an issue with older adults especially for exercise. Again, daily logbooks will be used during hospital stay, and during six weeks to register self-training conducted at home in intervention 1 but not intervention 2? • Description of the interventions needs to be clearer. Intervention 2 needs to be also conducted at home to answer your research question. It does not really make sense that such a comprehensive exercise intervention to be conducted only inpatient and not planned for continuity when discharged home. The patient can stay as short as 2 days to months, depending on the condition. Patient and public involvement  • While involving healthcare managers may help with the implementation of the intervention in the participating hospitals, the study protocol did not clearly describe how the implementation data will be collected beyond the use of the diaries. Suggest using the RE-AIM framework if implementation is part of the trial. Suggest also to label the trial: if it's an effectiveness-implementation, that would be a type II hybrid RCT. If it's a primarily effectiveness with some implementation, it would be a type I hybrid RCT. Since clinical effectiveness of exercise on physical function is well-explored in other literature, a hybrid type II trial may be more suitable. • Please describe the qualitative interview in brief. This can be a separate section. Overall, this article will benefit from professional English editing. For example, under the trial design section, "randomization will be done by the responsible researchers" has grammatical errors.
--	---

REVIEWER	Sáez de Asteasu, Mikel L Public University of Navarre
REVIEW RETURNED	03-Apr-2023

GENERAL COMMENTS	An interesting three-armed RCT protocol to assess the effectiveness of two different exercise programs compared to the hospital usual-care in acutely hospitalized older adults. Changes in functional capacity, assessed by the SPPB, and other health related endpoints including sarcopenia, falls and health related quality of life will be measured during hospitalization and post-discharge. The vulnerable population used is one of the potential strengths of this study. However, there are minor points for considering in the manuscript: Why are objectives included in the Methods section instead of the Introductions last section? Regarding the inclusion criteria, how will be severe confusion defined or evaluated? The exercise programs will be carried out at weekends? Martinez-Velilla et al included weekends as training days in their study. Participants with dementia are included in the study? The cognitive function of the older adults will be measured at admission?
---

	Finally, I kindly advise authors to have a look at the Vivifrail exercise program (www.vivifrail.com) for prescribing resistance exercises using weight cuffs adapted to the patient's functional capacity. Maybe could be useful for the intervention.
--	---

VERSION 1 – AUTHOR RESPONSE

Reviewer: 1

Thank you for sharing this protocol for your trial. I understand from the methods that recruitment and data collection commenced in September 2022. The protocol serves to answer an important knowledge deficit and is strengthened by a range of outcome variables. However, I would like the author to consider and address the following revisions:

1. The introduction should reference and highlight key findings from the Cochrane review published in November 2022 "Exercise for acutely hospitalised older medical patients". The authors commentary regarding implications for future research arising from the review, should serve as the justification for this trial.

Response: We thank the Reviewer for this suggestion. We have added the following text to the Introduction (page 5): *A recent Cochrane review concluded that there is uncertainty regarding the effect of exercise interventions on functional outcomes for older medical inpatients during acute hospitalization (9). Thus, it has been suggested that future research on this topic should focus on consistent reporting of participant characteristics, including baseline level of functional ability, as well as exercise dose and adherence to provide an insight into the reasons for the observed inconsistencies in findings (9).*

2. Terminology: the term older adults who are frail cited throughout the paper should be replaced with "older adults living with frailty". Inconsistent language noted throughout paper i.e trial, study, project. Please amend the following sentence as the language used, may be perceived as ageist "Acutely hospitalized patients aged 75 years and older often suffer from frailty and multimorbidity and consume a large proportion of the resources of health care (24)"

Response: The term "older adults who are frail" have been replaced with "older adults living with frailty". The words study and project have been replaced by trial throughout the manuscript. We have changed the above-mentioned sentence in the discussion (page 16) as follows: *Older adults consume a large proportion of health care resources (27).*

3. Eligible participants and baseline characteristics: While the authors have outlined inclusion and exclusion criteria for participation, please outline what clinical criteria or measurement will be utilised to determine "severe confusion". Inclusion of socioeconomic variables e.g level of education, previous occupation and a profile of baseline characteristics e.g. Charlson Comorbidity Index should be considered. Use of ICD 10 codes should be considered to standardise recording of "reason for hospital admission".

Response: To clarify the inclusion criteria, we have added the following text at page 7: *People who cannot follow instructions will be excluded. Patients who by the responsible physician are assessed as not eligible to participate due to terminal illness or any major medical condition that contraindicates exercise will also be excluded. We will also exclude patients who live in nursing homes, or those who were previously included in the trial.* Concerning socioeconomic variables, since the trial has

already started, we are not able to add new variables. We however collect data on cohabitation status, since living alone may be associated with being less physically active (e.g., Dohrn et al. 2020). Furthermore, we have added the following sentence to page 11: *Diseasediagnoses (reason for hospital admission and comorbidities) will be coded based on the International Classification of Diseases Tenth Revision (ICD-10), and further operationalized using a previously described methodology (22).*

4. Randomisation process: Given the risk of selection bias imposed by using the block randomisation design outlined in this protocol, please provide a further an evidence-based justification for using this design.

Response: We thank the Reviewer for her comment. We have added references for the block design at page 7. Furthermore, we have extended the discussion about the block design on page 17 as follows: *First, instead of a classic randomization we have chosen a block design, which may introduce selection bias in the trial. However, an advantage of the block design is that it avoids co-presence of participants from both groups at the same department; and thus, prevents that the exercises in the intervention groups will spread to ordinary treatment and risk to contaminate the control group. If the exercises in the intervention groups would spread to the control group and produce unexpected clinical benefits, it could create a bias by disguising a possible genuine benefit of the interventions and reduce the effect size to a value that is not clinically significant. Therefore, it has been emphasized that researchers in clinical trials on rehabilitation topics should make sure that control groups do not include any of the active elements of the intervention such as physical exercise (12).*

5. PPI: The PPI element of this protocol appears to be largely focused on healthcare professionals and researchers experiences of an interdisciplinary approach and less about the older adults experiences of recruitment and adherence to the intervention. Therefore, please provide further information on the planned conduct of the "qualitative interviews on the participants' views of the interventions".

Response: We have added the following sentence at page 14-15: *Furthermore, qualitative interviews will be conducted among the participants in the two intervention groups, to explore their experiences of the interventions. The interviews will be conducted at the hospital during the hospital stay, or at the participant's home after discharge.*

Reviewer: 2

Thank you for the opportunity to review your protocol paper, this area of research is very important. Thank you for furthering the evidence in this field. My comments are purely from a research and practice point of view and are not intended to malign any organization, or individual.

Strength and limitations of this study

1. Study participants will.....so the that the sample will be representative of older hospitalised patients in Sweden

Response: The sentence has been clarified as follows: *In order for the sample to be representative of older hospitalized patients, the participants will be consecutively recruited at several hospitals in Stockholm.*

2. While the project has a clear practice-oriented focus, it is not clear that it is interdisciplinary as only exercise (no nutrition or psychological) interventions were tested.

Response: We agree with the Reviewer that this was unclear. We have removed this sentence from the Strength and limitations of this study. Furthermore, we have clarified this issue at page 14 as follows: *Although the interventions will mainly be led by physiotherapists, an intervention at a geriatric*

department requires a dialog with several professions to meet the needs of the older adults, who often have several concurrent health problems. Thus, the trial has an interdisciplinary and practice-oriented focus, which will facilitate future implementation if the interventions are proven to be effective.

3. It is known that such interventions cannot be double blinded unless there is a sham intervention. As such, I do not think that there is a need to list that as a key limitation of the study in the key highlights.

Response: We thank the Reviewer for this suggestion. We have removed this sentence from the Strength and limitations of this study.

Introduction

4. Acutely hospitalized older patients are often frail... (as opposed to have frailty)

Response: The term has been rephrased as suggested (page 5).

5. Such periods of physical activity, albeit short, may exacerbate.....

Response: The sentence has been rephrased as suggested (page 5).

6. Please replace “ Identifying the most efficient interventions to reduce health consequences of acute hospitalization in older adults should be a high priority in society, especially the increasing older population” with something more specific to the acute care setting. I do get the message that there is ageing population but this sentence is too generic and does not highlight the importance of this study. Suggest to authors to highlight the time constraints in an acute setting and how a simple exercise, if proven effective, can be cost-effective in a time-pressure acute care setting.

Response: We thank the Reviewer for this suggestion. We have added the following text to the introduction (page 6): *Considering the time constraints in acute care settings, a simple exercise program may be more feasible if proven effective.*

Methods and analysis

7. The study has been registered at clinicaltrials.gov. à Please provide the registration details i.e., number.

Response: The registration number has been added on page 7.

8. Please reference appropriately. For example, the SPRIT checklist needs to be referenced, along with previous studies that used block design.

Response: We have added references for the SPRIT checklist and block design at page 7. Furthermore, we have extended the discussion about the block design on page 17-18 as follows: *First, instead of a classic randomization we have chosen a block design, which may introduce selection bias in the trial. However, an advantage of the block design is that it avoids co-presence of participants from both groups at the same department; and thus, prevents that the exercises in the intervention groups will spread to ordinary treatment and risk to contaminate the control group. If the exercises in the intervention groups would spread to the control group and produce unexpected clinical benefits, it could create a bias by disguising a possible genuine benefit of the interventions and reduce the effect size to a value that is not clinically significant. Therefore, it has been emphasized that researchers in clinical trials on rehabilitation topics should make sure that control groups do not include any of the active elements of the intervention such as physical exercise(12).*

9. Please provide details on how usual care will be documented. We'd expect varied care depending on physician's referral to physiotherapy departments during the stay.

Response: We thank the Reviewer for the opportunity to clarify this issue. We have added the following text in the Methods (page 8): *The control group will receive usual care. Usual care is based on teamwork, which includes physiotherapy. The physiotherapy focuses on mobility assessments, and the patients may receive a limited amount of functional training when needed. However, we expect that the participants often have a limited amount of physical activity during the hospital stay (15).* Furthermore, we have added the following text in the Discussion (page 18): *Second, like usual care in the studies by Martinez-Velilla et al. and Ortiz-Alonso et al. (10, 11), usual care at geriatric departments in Sweden include physiotherapy, and the patients may receive a limited amount of functional training when needed. This may introduce a bias in the trial since participants in the control group may also receive some physical exercise. However, a hospital stay is often accompanied by low levels of physical activity, which has been suggested to play a major role in causing the negative health consequences associated with hospitalization (4, 15).*

10. The mini-nutritional assessment tool short form is a screening tool (assess nutritional risk) and cannot provide nutrition assessment. Please change the description where this is discussed within the manuscript.

Response: We have changed nutritional status to nutritional risk throughout the manuscript.

11. Nutrition supplementation is collected during the hospitalization. Any physical activity in standard care should also be documented too because standard care varies between hospitals

Response: Please see our response to Question 9 above. The trial has already started. When planning the trial, we realized that it would not be possible to reliably document the amount of physical activity in standard care. Similarly, to the studies by Martinez-Velilla et al. and Ortiz-Alonso et al, we describe usual care but do not collect data on the amount of physical activity included.

Discussion

12. Patients in intervention 1 will be encouraged to continue with the exercises at home but not intervention 2?

Response: We thank the Reviewer for the opportunity to clarify this issue. We have added the following text in the Discussion (page 17): *We expect that this trial will help identify the exercise program that is most effective and therefore may be most suitable for future implementation. The two exercise programs have different possible strengths and weaknesses. For instance, it may be hypothesized that the comprehensive exercise program may have a better effect on physical function at hospital discharge since it is more individualized and intensive than the simple exercise program. However, a disadvantage is that it is less suitable for self-training than the simple exercise program, due to its complexity and the possible risk entailed by doing unsupervised balance training. Thus, it may be hypothesized that the simple exercise program, which is more suitable for self-training, can have an equal or better effect on physical function than the comprehensive exercise program three months after discharge since the participants can continue with the program at home. Furthermore, in future studies we plan to conduct post-hoc analyses to identify associations between baseline factors, e.g., frailty level, and the effect of the intervention.*

13. While I agree that measuring effectiveness is important, we know that compliance is an issue with older adults especially for exercise. Again, daily logbooks will be used

during hospital stay, and during six weeks to register self-training conducted at home in intervention 1 but not intervention 2?

Response: Please see our response to Question 12 above.

14. Description of the interventions needs to be clearer. Intervention 2 needs to be also conducted at home to answer your research question. It does not really make sense that such a comprehensive exercise intervention to be conducted only inpatient and not planned for continuity when discharged home. The patient can stay as short as 2 days to months, depending on the condition.

Response: The purpose of these interventions is to mitigate the negative health consequences associated with hospitalization among older adults. Therefore, the duration of the interventions at the hospital will be as long as the hospital stay. We have added a sentence to clarify this issue in the Discussion (page 18): *However, a hospital stay is often accompanied by low levels of physical activity, which has been suggested to play a major role in causing the negative health consequences associated with hospitalization (4, 15).* Please also see our response to Question 12 above.

Patient and public involvement

15. While involving healthcare managers may help with the implementation of the intervention in the participating hospitals, the study protocol did not clearly describe how the implementation data will be collected beyond the use of the diaries. Suggest using the RE-AIM framework if implementation is part of the trial. Suggest also to label the trial: if it's an effectiveness-implementation, that would be a type II hybrid RCT. If it's a primarily effectiveness with some implementation, it would be a type 1 hybrid RCT. Since clinical effectiveness of exercise on physical function is well-explored in other literature, a hybrid type II trial may be more suitable.

Response: We thank the Reviewer for the opportunity to clarify this issue. Implementation is not part of the trial in its initial phase. We have clarified this by adding the following sentence to the Discussion (page 16): *If the trial will show positive effects, we plan to further study its implementation and analyze its cost-effectiveness in subsequent studies.*

16. Please describe the qualitative interview in brief. This can be a separate section.

Response: We have added the following text at page 14-15: *Furthermore, qualitative interviews will be conducted among the participants in the two intervention groups, to explore their experiences of the interventions. The interviews will be conducted at the hospital during the hospital stay, or at the participant's home after discharge.* We think that the methodology of these interviews is beyond the scope of this protocol.

17. Overall, this article will benefit from professional English editing. For example, under the trial design section, "randomization will be done by the responsible researchers" has grammatical errors.

Response: The article has undergone English editing.

Reviewer: 3

An interesting three-armed RCT protocol to assess the effectiveness of two different exercise programs compared to the hospital usual-care in acutely hospitalized older adults. Changes in functional capacity, assessed by the SPPB, and other health related endpoints including sarcopenia, falls and health related quality of life will be measured during hospitalization and post-discharge. The

vulnerable population used is one of the potential strengths of this study. However, there are minor points for considering in the manuscript:

1. Why are objectives included in the Methods section instead of the Introductions last section?

Response: We have moved the objectives to the Introduction.

2. Regarding the inclusion criteria, how will be severe confusion defined or evaluated?

Response: We have clarified the inclusion criteria at page 7: *Inclusion criteria for participation are age 75 years and older, ability to stand up from a sitting position independently or with minimal personal help, and ability to communicate and collaborate with the research staff. People who cannot follow instructions will be excluded. Patients who by the responsible physician are assessed as not eligible to participate due to terminal illness or any major medical condition that contraindicates exercise will also be excluded. We will also exclude patients who live in nursing homes, or those who were previously included in the trial.*

3. The exercise programs will be carried out at weekends? Martinez-Velilla et al included weekends as training days in their study.

Response: We thank the Reviewer for the opportunity to clarify this issue. We have added the following text in the Discussion (page 17): *Finally, the interventions will only be performed on weekdays to fit with staff resources during weekends.*

4. Participants with dementia are included in the study?

Response: Yes, people with dementia may be included if they fulfil the inclusion criteria. Please also see our response to Question 2 above.

5. The cognitive function of the older adults will be measured at admission?

Response: Cognition is not included in the assessment battery, although neuropsychological problems is included as one of the items in Mini Nutritional Assessment-Short Form.

6. Finally, I kindly advise authors to have a look at the Vivifrail exercise program (<https://eur01.safelinks.protection.outlook.com/?url=http%3A%2F%2Fwww.vivifrail.com%2F&data=05%7C01%7Canna-karin.welmer%40ki.se%7C0bbd1616c352450dbcc808db41a096f8%7Cbff7eef1cf4b4f32be3da1dda043c05d%7C0%7C0%7C638175930204495377%7CUnknown%7CTWFpbGZsb3d8eyJWljojMC4wLjAwMDAiLCJQIjoiV2luMzliLCJBTil6lk1haWwiLCJXVC16Mn0%3D%7C3000%7C%7C%7C&sdata=1FrQXQZiFIYEp8OEMZ2iVrq7gyQcNkeP06h151z0mM8%3D&reserved=0>) for prescribing resistance exercises using weight cuffs adapted to the patient's functional capacity. Maybe could be useful for the intervention.

Response: We thank the Reviewer for this suggestion. Since the trial has already started, we however have limited possibilities to adjust the interventions.

VERSION 2 – REVIEW

REVIEWER	Han, chad Yixian Flinders University College of Nursing and Health Sciences
REVIEW RETURNED	23-May-2023

GENERAL COMMENTS	The authors have addressed my comments adequately. I have no further comments.
--

REVIEWER	Sáez de Asteasu, Mikel L Public University of Navarre
REVIEW RETURNED	05-Jun-2023

GENERAL COMMENTS	Thank you. The authors have addressed all the reviewer's suggestions and comments.
--